# *TERT* Promoter Mutation and Extent of Thyroidectomy in Patients with 1–4 cm Intrathyroidal Papillary Carcinoma

**DOI:** 10.3390/cancers12082115

**Published:** 2020-07-30

**Authors:** Aya Ebina, Yuki Togashi, Satoko Baba, Yukiko Sato, Seiji Sakata, Masashi Ishikawa, Hiroki Mitani, Kengo Takeuchi, Iwao Sugitani

**Affiliations:** 1Department of Endocrine Surgery, Nippon Medical School, Tokyo 113-8603, Japan; isugitani@nms.ac.jp; 2Pathology Project for Molecular Targets, The Cancer Institute, Japanese Foundation for Cancer Research, Tokyo 135-8550, Japan; yuki.togashi@jfcr.or.jp (Y.T.); satoko.baba@jfcr.or.jp (S.B.); yukiko.sato@jfcr.or.jp (Y.S.); sakata-fjt@umin.ac.jp (S.S.); kentakeuchi-tky@umin.net (K.T.); 3Division of Pathology, The Cancer Institute, Japanese Foundation for Cancer Research, Tokyo 135-8550, Japan; 4Department of Anesthesiology, Nippon Medical School, Tokyo 113-8603, Japan; masashi-i@nms.ac.jp; 5Division of Head and Neck, Cancer Institute Hospital, Tokyo 135-8550, Japan; mitani@jfcr.or.jp; 6Clinical Pathology Center, The Cancer Institute Hospital, Japanese Foundation for Cancer Research, Tokyo 135-8550, Japan

**Keywords:** papillary thyroid carcinoma, *BRAF V600E*, *TERT* promoter mutations, lobectomy, total thyroidectomy

## Abstract

There are concerns regarding overtreatment in papillary thyroid carcinoma (PTC). *BRAF V600E* and *TERT* promoter mutations play important roles in the development of PTC. However, initial surgical approaches for PTC based on genetic characteristics remain unclear. The present study aimed to identify genetic mutations as predictors of prognosis and to establish proper indications for lobectomy (LT) in patients with 1–4 cm intrathyroidal PTC. Prospectively accumulated data from 685 consecutive patients with PTC who underwent primary thyroid surgery at the Cancer Institute Hospital, Tokyo, Japan, between 2001 and 2012 were retrospectively reviewed. Of the 685 patients examined, 538 (78.5%) had *BRAF V600E* mutation and 133 (19.4%) had *TERT* promoter mutations. Patients with *TERT* promoter mutations displayed significantly worse outcomes than those without mutations (10-year cause-specific survival (CSS): 73.7% vs. 98.1%, *p* < 0.001; 10-year disease-free survival (DFS): 53.7% vs. 93.3%, *p* < 0.001). As for extent of thyroidectomy among *TERT* mutation-negative patients with 1–4 cm intrathyroidal PTC, patients who underwent LT showed no significant differences in 10-year CSS and 10-year DFS compared to patients who had total thyroidectomy (TT) under propensity score-matching. Avoiding TT for those patients indicates a possible pathway to prevent overtreatment and reduce postoperative complications.

## 1. Introduction

The incidence of thyroid cancer has been increasing dramatically in developed countries. Most cases involve low-risk papillary thyroid carcinoma (PTC), showing favorable treatment outcomes. Thus, mortality due to thyroid cancer has remained stable and debate is growing regarding the overdiagnosis and overtreatment of thyroid cancers [1]. Traditionally, total thyroidectomy (TT) had been the standard surgical procedure for PTC in Western countries, whereas lobectomy (LT) to preserve postoperative thyroid function has been widely performed in Japan. Recently, risk-adapted management has been considered as a cornerstone to the treatment of patients with differentiated thyroid carcinoma. This policy determines the therapeutic strategy, including the extent of thyroidectomy and postoperative adjuvant therapies based on the risks of cancer recurrence and mortality [2]. The 2015 American Thyroid Association (ATA) guidelines recommended either TT or LT as the initial surgical approach for patients with 1–4 cm intrathyroidal PTC [3]. They acknowledged that LT alone might be sufficient for low-risk PTC. Indeed, the rate of patients who undergo LT has since been increasing [4]; however, the majority of patients with PTC in the United States still undergo TT [5]. This might be partially due to a lack of decisive factors to determine indications for LT among such patients.

Several studies have investigated the genetic molecular profiles of PTC and revealed *BRAF V600E* mutation and *TERT* promoter mutation as major mutations playing important roles in PTC development [6,7]. Although some studies have examined relationships between those mutations and prognosis, no studies have clarified the initial surgical approach for patients with PTC based on genetic characteristics. The aim of the present study was to identify mutations as predictors of prognosis and to suggest appropriate indications for LT in patients with 1–4 cm intrathyroidal PTC using the genetic status of the tumor.

## 2. Results

Table 1 shows baseline characteristics of the 685 patients. Mean duration of follow-up after initial surgery was 10 ± 3 years. Mean age was 52 ± 14 years. The proportion of women was 75.9% (520 cases).

### 2.1. Gene Mutations and Outcome

Among the 685 patients, 538 (78.5%) had *BRAF V600E* mutation and 133 (19.4%) had *TERT* promoter mutations (C228T-positive, 112 cases; C250T-positive, 21 cases). Kaplan–Meier survival analysis showed that *BRAF V600E* mutations had no significant effect on 10-year cause-specific survival (CSS) (positive vs. negative: 93.1% vs. 93.6%, respectively; *p* = 0.998) and 10-year disease-free survival (DFS) (86.0% vs. 88.2%, respectively; *p* = 0.872) (Figure 1). In contrast, patients with *TERT* promoter mutations (TERT+) displayed significantly worse outcomes than those without *TERT* promoter mutations (TERT−) (10-year CSS: 73.7% vs. 98.1%, *p* < 0.001; 10-year DFS: 53.7% vs. 93.3%, *p* < 0.001, respectively) (Figure 2). As shown in Figure 3, patients with coexisting *BRAF V600E* and *TERT* promoter mutations (B+/T+) showed worse CSS and DFS than those with neither mutation (B−/T−) (10-year CSS: 75.3% vs. 96.4%, *p* < 0.001; 10-year DFS: 55.5% vs. 90.9%, *p* < 0.001). Only 7 patients displayed *TERT* promoter mutation alone (B−/T+), but these patients also showed significantly worse outcomes compared to patients with B−/T− (10-year CSS: 35.7%, *p* < 0.001; 7-year DFS: 20.0%, *p* < 0.001). On the other hand, *BRAF V600E* mutation alone (B+/T−) was not significantly poorer than patients with B−/T− (10-year CSS: 98.7%, *p* = 0.021; 10-year DFS: 94.1%, *p* = 0.086). Multivariate Cox proportional hazard regression analysis for genetic mutation status revealed that B+/T+ and B−/T+ were independently associated with poor cause-specific survival (Table 2).

The characteristics of patients with/without *BRAF V600E* mutation and with/without *TERT* promoter mutation are shown in Table 3 and Table 4, respectively. *BRAF V600E* mutation was significantly more common among older patients (*p* < 0.001) and cases with higher T and stage at diagnosis (*p* < 0.001); however, it was inversely associated with higher N and M. *TERT* promoter mutations were significantly more common among older patients (*p* < 0.001), men (*p* < 0.001), and cases with higher stage at diagnosis (*p* < 0.001). Multivariate Cox proportional hazard regression analysis showed that *TERT* promoter mutations (HR = 5.23, 95% CI 2.33–12.78, *p* < 0.001), age ≥55 years (HR = 2.47, 95% CI 1.09–6.38, *p* = 0.030), tumor size >4 cm (HR = 3.90, 95% CI 1.89–8.42, *p* < 0.001), N1b (HR = 3.27, 95% CI 1.30–8.82, *p* = 0.011) and M1 (HR = 2.46, 95% CI 1.17–5.15, *p* = 0.018) were independent risk factors associated with mortality (Table 5).

### 2.2. Extent of Thyroidectomy and TERT Promoter Mutations for Patients with 1–4 cm Intrathyroidal PTC

A total of 309 patients had intrathyroidal PTC with a maximal diameter of 1–4 cm without extrathyroidal extension, clinical evidence of any lymph node metastasis and distant sites. Here, we included T3b tumor invading only the strap muscles into intrathyroidal PTC. We investigated the relationship between extent of thyroidectomy and outcomes according to positivity for *TERT* promoter mutations (TERT+/TERT−) in this patient group, which comprised 33 TERT+ patients and 276 TERT− patients. Among TERT− patients, 59 patients underwent TT and 217 patients received LT as the initial thyroidectomy. Table 6 shows patient characteristics in the TERT− group according to the extent of thyroidectomy. Patients who underwent TT were significantly older, presenting with a higher proportion of T3b tumors and bilateral disease compared to LT patients. We thus conducted propensity score-matching to compare treatment outcomes between patients who underwent TT and LT. Before matching, no significant differences were evident in 10-year CSS (LT vs. TT: 100% each, *p* = 0.575) or 10-year DFS (97.4% vs. 96.9%, *p* = 0.773) (Figure 4). Likewise, no significant differences were seen in 10-year CSS (LT vs. TT; 100% each, *p* = 0.308) or 10-year DFS (96.6% vs. 96.9%, *p* = 0.554) when comparing LT to TT after matching (Figure 5). Incidences of temporary and permanent postoperative hypoparathyroidism in TT were significantly higher than those in LT. Five patients showed recurrence after LT in the TERT− group, comprising cervical lymph node recurrence in 4 cases and distant recurrence in the remaining 1 case. All patients with cervical recurrence were able to be cured by salvage surgery. One female patient who developed lung metastasis 5 years after the first operation did not undergo remnant thyroid resection and radioactive iodine (RAI) therapy because she was 85 years old at that time. She died 10.4 years after the first operation.

As for the TERT+ group, no significant differences in 10-year CSS (100% each, not significant) was seen comparing LT to TT. However, the 10-year DFS of patients treated by LT tended to be worse than that of TT (64.5% vs. 100%, *p* = 0.094) (Figure 6).

## 3. Discussion

Previously, a “one-size-fits-all” policy was the mainstream when determining the initial treatment procedure for patients with PTC. This took the form of TT and radioactive iodine (RAI) in Western countries, and LT in Japan. However, risk-adapted management policies based on appropriate risk stratification systems have been widely adopted more recently.

The 2015 ATA guidelines have expanded the indications for LT based on evidence that T1 or T2N0M0 can provide good long-term prognosis [8,9], and either TT or LT can be safely indicated for patients with 1–4 cm PTC without gross extrathyroidal extension, lymph node metastasis (LNM) or distant metastasis [3]. As a result, the rate of LT for thyroid cancer increased significantly from 17.3% to 22.0% in the United States after the release of the guidelines [4]. However, treatment teams are still apt to choose TT to enable RAI therapy or enhance follow-up based upon disease features and/or patient preferences. Indeed, Welch et al. reported that about 80% of patients with localized PTC ≤ 2 cm underwent TT [5]. They suggested that further efforts to reduce overtreatment by TT are needed, because TT is associated with a significantly higher risk of complications like hypoparathyroidism and recurrent laryngeal nerve palsy compared to LT, even among high-volume surgeons [10]. Moreover, all patients who undergo TT require lifelong thyroid hormone-replacement therapy.

On the other hand, LT has been the preferred operative approach for the majority of patients with PTC in Japan [11,12]. We designed our own risk group classification for predicting cause-specific death from PTC and have recommended LT as the treatment option for low-risk patients [13]. The risk-group definition described in the Materials and Methods section featured a wider range for the low-risk group compared to other risk-group stratification systems. We reported the validity of the definition in 2014 and showed that the 10-year CSS for patients with low-risk PTC was 99% and did not differ between patients who underwent TT versus LT [14]. However, cause-specific mortality (1%) and tumor recurrence (8.3%) were seen even within the low-risk group. We therefore attempted to identify a decisive genetic marker to predict prognosis of PTC and to determine the initial treatment procedure.

Several genetic alterations have been identified in PTC, mainly involving genes of the MAP kinase pathway (*BRAF* and *RAS* point mutations). Although *BRAF V600E* mutation is the most prevalent point mutation in PTC, the contribution of *BRAF V600E* mutation to PTC outcomes remains controversial [6,15,16,17,18,19]. In Japan, Ito et al. reported a relatively high prevalence (38.4%) of *BRAF* mutation among 631 patients with PTC, but the mutation did not correlate with high-risk features or DFS [19]. Indeed, we also found a high incidence (78.5%) of *BRAF V600E* mutation and no impact of this mutation on CSS and DFS in this Japanese series. Further study would be needed to clarify the reason why Japanese patients showed higher incidence of *BRAF* mutation than other countries.

*TERT* promoter mutations were found in 3.5–14.0% of PTC [20,21,22], mostly comprising C228T mutations. They are known as effective risk factors for patients with PTC. Several investigators have shown that *TERT* promoter mutations are significantly more common among older patients, bigger tumor size, more aggressive subtype, tumors with extrathyroidal invasion, distant metastasis or higher stage at diagnosis, and are related to poor outcomes [7,23,24,25,26,27,28]. In addition, Xing et al. reported that coexisting *BRAF V600E* and *TERT* promoter mutations represented a strong predictor for the most aggressive PTCs with the highest recurrence rate [29]. Some meta-analyses have verified that coexistence of both mutations has a synergistic effect on aggressive clinicopathological characteristics and even cancer-related mortality for patients with PTC [30,31]. In this series, the incidence of *TERT* promoter mutations was 19.4% and we revealed that *TERT* promoter mutations alone and coexisting *BRAF V600E* and *TERT* promoter mutations were independently related to CSS for patients with PTC. However, most (126 of 133) patients with *TERT* promoter mutations also had *BRAF* mutation and only 7 patients showed *TERT* mutations alone. Consequently, *BRAF V600E* mutation was not a better indicator associated with CSS and DFS compared to *TERT* mutations in this population. Thus, we simply applied *TERT* promoter mutation to examine the relationship between extent of thyroidectomy and outcomes for patients with 1–4 cm intrathyroidal PTC. We revealed that LT could provide favorable outcomes identical to TT for those patients without *TERT* promoter mutations. The recent advent of molecular tests using fine-needle aspiration cytology specimens has enabled preoperative diagnosis of the existence of *TERT* promoter mutations [32]. Therefore, surgeons can decide the initial surgical approach for patients with PTC based on the status of this genetic marker.

The strength of our study lies in the large cohort treated under a uniform strategy and accompanied by long-term, precise outcome information. Conversely, the study was a retrospective analysis of prospectively accumulated data from a single, tertiary oncology referral center in Japan, where RAI therapy had not been utilized much compared to Western countries. Global, prospective verification involving multiple institutions is expected in the near future. In addition, our series included only five patients under 18 years old (2 were positive for *BRAF* mutation but no one had *TERT* mutations). Thus, the conclusions provided should be applied only for adult patients so far.

## 4. Materials and Methods

### 4.1. Study Design and Patients

After approval by the institutional review board (IRB number 2013–1128, 24 January 2014), we retrospectively reviewed the prospectively accumulated data of 685 consecutive patients with PTC who underwent primary thyroid surgery at the Cancer Institute Hospital, Tokyo, Japan, between 2001 and 2012. All subjects gave their informed consent for inclusion before they participated in the study. Patients with papillary microcarcinoma (maximal diameter, ≤1.0 cm) were excluded. Patients were classified preoperatively into low- and high-risk groups according to our risk group classification system [13]. That is, patients with distant metastasis and older patients (≥50 years) with massive extrathyroidal invasion or large LNM (maximal diameter, ≥3 cm) were defined as high risk and all other patients were classified as low risk. Although we presented the treatment options (including both LT and TT) to establish shared decision-making for patients with low-risk PTC, we basically recommended LT when the tumor was unilateral. Radioactive iodine (RAI) treatment was usually performed for patients with high-risk features [14]. Extent of lymph node dissection was determined under the previously reported principle [33]: a) Dissection of the central compartment (level VI) alone for patients with LNM only in the central zone or with no LNM; and b) lateral neck dissection (basically, levels II, III, IV, and VI) when the patient was diagnosed with lateral neck LNM. Bilateral neck dissection was performed only when preoperative imaging showed bilateral neck LNM. Postoperative surveillance to evaluate recurrent lesions was conducted every 6 or 12 months, using cervical ultrasound, chest X-rays or computed tomography. Patients with distant metastasis at the time of initial presentation were excluded from the analysis of recurrence or DFS.

### 4.2. DNA Extraction and Mutation Screening

Genomic DNA was extracted using a DNeasy Blood & Tissue Kit (QIAGEN, Hilden, Germany) from fresh frozen specimens. To analyze the presence of *BRAF V600E* mutation and *TERT* promoters C228T and C250T, multiplex genomic PCR was performed with the primers at the adjusted concentrations shown in Table 1 using PrimeSTAR^®^ GXL DNA Polymerase (TaKaRa Bio, Shiga, Japan). The template genomic DNA was subjected to 35 cycles of denaturation at 98 °C for 10 s, annealing at 60 °C for 15 s, and polymerization at 68 °C for 15 s. The PCR product was separated by agarose gel electrophoresis. After purification with Sephadex G-75 (GE Healthcare, Buckinghamshire, UK), the amplified PCR product was used for the single-base extension reaction, using a SNaPshot^®^ Multiplex System (Thermo Fisher Scientific, Waltham, MA, USA) according to the instructions from the manufacturer. Concentrations of primers were optimized as described in Table 7. To confirm the results from SNaPshot, Sanger sequencing analysis of the *BRAF V600E* mutation and *TERT* promoters C228T and C250T were also performed on some cases.

### 4.3. Digital PCR for TERT Promoter Mutations

To confirm the presence of *TERT* promoters C228T and C250T, digital PCR using a QuantStudio™ 3D Digital PCR System (Thermo Fisher Scientific) was also performed in some cases. Mutation analysis was conducted with TaqMan Liquid Biopsy dPCR Assays (Thermo Fisher Scientific) which were specifically designed for *TERT* promoters C228T and C250T (Hs000000092_rm and Hs000000093_rm, respectively).

### 4.4. Statistical Analysis

Clinical data were recorded and tabulated in Excel software (Microsoft, Redmond, WA, USA). All analyses were performed using JMP for Windows v11.0.0 (SAS Institute, Cary, NC, USA). Values of *p* < 0.05 were considered significant. Results are expressed in the form of mean ± standard deviation or *n* (%). Clinical characteristics were compared between groups using the chi-square test for categorical variables and Mann–Whitney’s U test for continuous variables. Survival curves were determined using the Kaplan–Meier method. Cox proportional hazard modeling and log-rank testing were used to compare time-to-event distributions. Using the propensity score-matching method, patients with or without genetic mutations related to prognosis who underwent LT and TT were matched by age, sex, tumor size, and extrathyroidal invasion of the strap muscles in a 1:1 ratio. After propensity score-matching, baseline characteristics were compared between the two groups as matched pairs. Kaplan–Meier curves were constructed, and the log-rank test was used to compare CSS and DFS.

## 5. Conclusions

In summary, we found that *TERT* promoter mutations were independently related to poor outcomes for patients with PTC. In contrast, *BRAF V600E* mutation alone was not significantly associated with aggressiveness. In addition, patients with 1–4 cm intrathyroidal PTC without *TERT* promoter mutations could obtain favorable outcomes from LT. The present findings suggest a pathway to prevent oversurgery and reduce postoperative complications for those patients, leading to alleviation of physical, psychological, and economic burdens on patients.

## Figures and Tables

**Figure 1 cancers-12-02115-f001:**
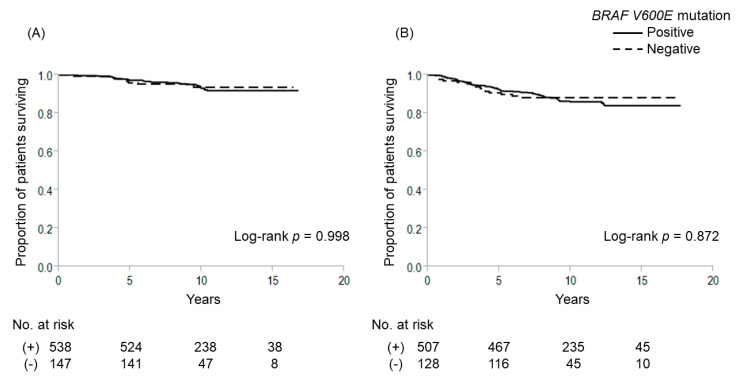
Cause-specific and disease-free survival curves for patients with or without *BRAF V600E* mutation. (**A**) Cause-specific survival; (**B**) disease-free survival.

**Figure 2 cancers-12-02115-f002:**
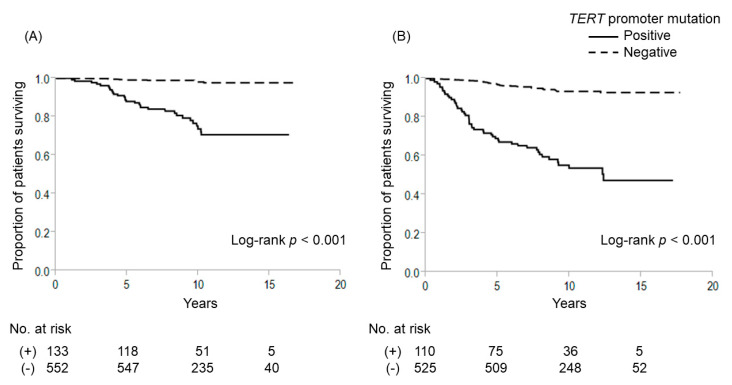
Cause-specific and disease-free survival curves for patients with or without *TERT* promoter mutations. (**A**) Cause-specific survival; (**B**) disease-free survival.

**Figure 3 cancers-12-02115-f003:**
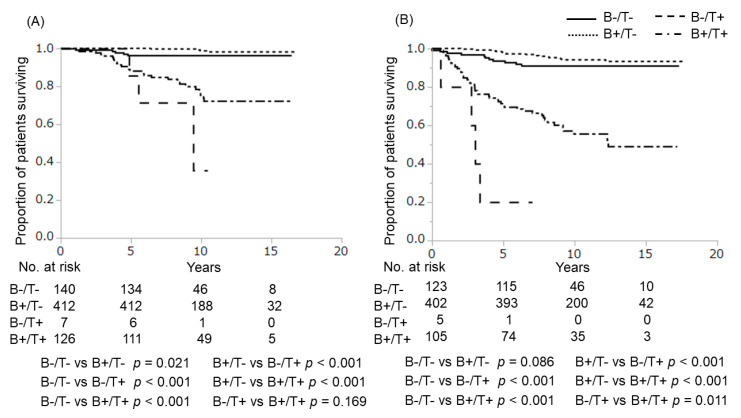
Cause-specific and disease-free survival curves for patients with *TERT* promoter mutations alone, *BRAF* mutation alone, both mutations and neither mutation. (**A**) Cause-specific survival; (**B**) disease-free survival. B+, *BRAF V600E* mutation-positive; B−, *BRAF V600E* mutation-negative; T+, *TERT* promoter mutation-positive; T−, *TERT* promoter mutation-negative.

**Figure 4 cancers-12-02115-f004:**
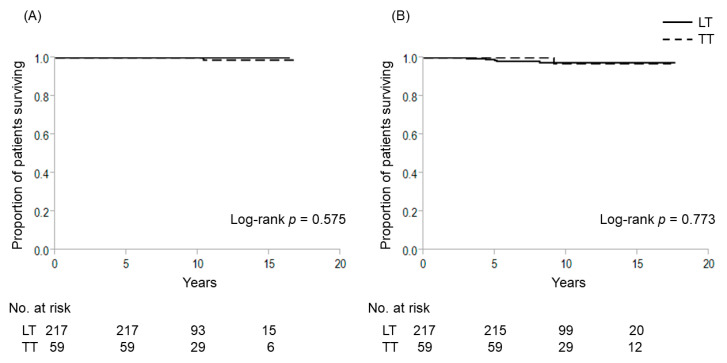
Cause-specific and disease-free survival curves for patients without *TERT* promoter mutations who underwent lobectomy and total thyroidectomy for patients with intrathyroidal PTC with a maximal diameter of 1–4 cm. (**A**) Cause-specific survival; (**B**) disease-free survival. LT, lobectomy; TT, total thyroidectomy.

**Figure 5 cancers-12-02115-f005:**
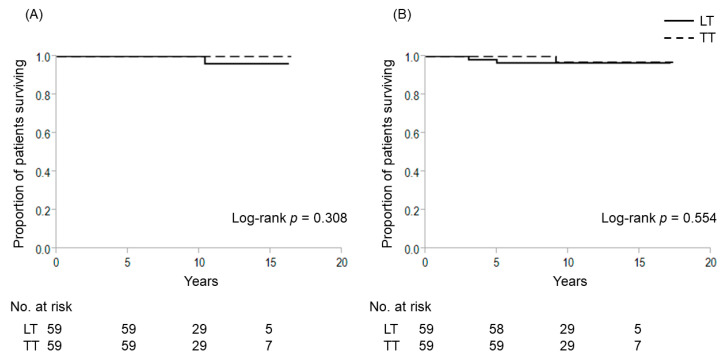
Cause-specific and disease-free survival curves for patients without *TERT* promoter mutations who underwent lobectomy and total thyroidectomy after propensity score-matching for patients with intrathyroidal PTC with a maximal diameter of 1–4 cm. (**A**) Cause-specific survival; (**B**) disease-free survival. LT, lobectomy; TT, total thyroidectomy.

**Figure 6 cancers-12-02115-f006:**
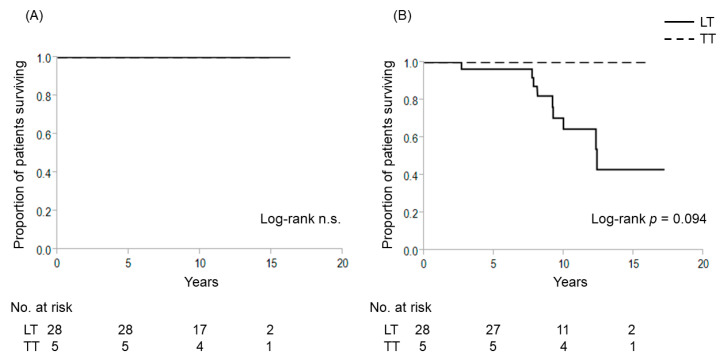
Cause-specific and disease-free survival curves for patients with *TERT* promoter mutations who underwent lobectomy and total thyroidectomy for patients with intrathyroidal PTC with a maximal diameter of 1–4 cm. (**A**) Cause-specific survival; (**B**) disease-free survival. n.s., non-significant, LT, lobectomy; TT, total thyroidectomy.

**Table 1 cancers-12-02115-t001:** Baseline characteristics of patients.

Characteristics	*n* = 685
Follow-up duration, years (range)	10 ± 3 (1–17)
Age, years (range)	52 ± 14 (15–86)
Age, *n* (%)	
≥55 years	322 (47.0%)
<55 years	363 (53.0%)
Sex, female, *n* (%)	520 (75.9%)
T, *n* (%) AJCC/UICC 8th edition	
1b	211 (30.8%)
2	82 (12.0%)
3a	20 (2.9%)
3b	229 (33.5%)
4a	142 (20.7%)
4b	1 (0.1%)
N, *n* (%) AJCC/UICC 8th edition	
0	362 (52.9%)
1a	87 (12.7%)
1b	236 (34.4%)
M, *n* (%) AJCC/UICC 8th edition	
0	635 (92.7%)
1	50 (7.3%)
Stage classification AJCC/UICC 8th edition, *n* (%)	
I	409 (59.6%)
II	170 (24.9%)
III	78 (11.4%)
IVA	0 (0%)
IVB	28 (4.1%)
Tumor size, *n* (%)	
<4 cm	583 (85.1%)
≥4 cm	102 (14.9%)
Total thyroidectomy, *n* (%)	225 (32.8%)
Radioactive iodine therapy, ≥30 mCi, *n* (%)	102 (14.9%)

**Table 2 cancers-12-02115-t002:** Cox proportional hazard regression analysis of genetic mutation status associated with outcomes.

Mutations	Cause-Specific Death			Recurrence		
	HR	95% CI	*p* Value	HR	95% CI	*p* Value
No mutations	1			1		
*BRAF V600E* mutation only	0.56	0.28–3.12	0.131	0.4	0.07–3.10	0.347
*TERT* promoter mutation only	22.83	6.25–67.91	<0.001	14.34	0.67–149.97	0.078
*BRAF V600E* + *TERT* promoter mutations	5.8	3.12–11.79	<0.001	9.18	2.63–57.91	<0.001

HR, hazard ratio; CI, confidence interval.

**Table 3 cancers-12-02115-t003:** Characteristics of patients with and without *BRAF V600E* mutation.

Characteristics	*BRAFV600E* Status, *n* (%)		
	Wild Type	Mutated	*p* Value
*N*	147	538	
Follow-up duration, years (range)	9 ± 3 (1–16)	10 ± 3 (1–17)	<0.001
Age, years (range)	46 ± 15 (15–81)	54 ± 14 (15–86)	<0.001
Age, *n* (%)			
≥55 years	44 (29.9%)	270 (50.2%)	<0.001
<55 years	103 (70.1%)	268 (49.8%)	
Sex, female, *n* (%)	103 (70.1%)	417 (77.5%)	0.066
T, *n* (%) AJCC/UICC 8th edition			
1	53 (36.1%)	158 (29.4%)	<0.001
2	30 (20.4%)	52 (9.7%)	
3a	9 (6.1%)	11 (2.0%)	
3b	34 (23.1%)	195 (36.2%)	
4a	21 (14.3%)	121 (22.5%)	
4b	0 (0%)	1 (0.2%)	
N, *n* (%) AJCC/UICC 8th edition			
0	70 (47.6%)	292 (54.3%)	0.045
1a	14 (9.5%)	73 (13.6%)	
1b	63 (42.9%)	173 (32.2%)	
M, *n* (%) AJCC/UICC 8th edition			
0	128 (87.1%)	507 (94.2%)	0.006
1	19 (12.9%)	31 (5.8%)	
Stage classification AJCC/UICC 8th edition, *n* (%)			
I	107 (72.8%)	302 (56.1%)	<0.001
II	24 (16.4%)	146 (27.1%)	
III	8 (5.4%)	70 (13.1%)	
IVA	0 (0%)	0 (0%)	
IVB	8 (5.4%)	20 (3.7%)	
Tumor size, *n* (%)			
<4 cm	120 (81.6%)	463 (86.1%)	0.191
≥4 cm	27 (18.4%)	75 (13.9%)	
Total thyroidectomy, *n* (%)	49 (33.3%)	176 (32.7%)	0.887
Radioactive iodine therapy, ≥30 mCi, *n* (%)	27 (18.4%)	75 (13.9%)	0.191

**Table 4 cancers-12-02115-t004:** Characteristics of patients with and without *TERT* promoter mutations.

Characteristics	*TERT* Status, *n* (%)		
	Wild Type	Mutated	*p* Value
*N*	552	133	
Follow-up duration, years (range)	12 ± 5 (1–25)	9 ± 3 (1–17)	0.03
Age, years (range)	52 ± 13 (15–86)	64 ± 10 (27–86)	<0.001
Age, *n* (%)			
≥55 years	214 (38.8%)	108 (81.2%)	<0.001
<55 years	338 (61.2%)	25 (18.8%)	
Sex, female, *n* (%)	435 (78.8%)	85 (63.9%)	<0.001
T, *n* (%) AJCC/UICC 8th edition			
1	206 (37.3%)	5 (3.8%)	<0.001
2	65 (11.8%)	17 (12.8%)	
3a	14 (2.6%)	6 (4.4%)	
3b	189 (34.2%)	40 (30.1%)	
4a	78 (14.1%)	64 (48.1%)	
4b	0 (0%)	1 (0.8%)	
N, *n* (%) AJCC/UICC 8th edition			
0	308 (55.8%)	54 (40.6%)	0.003
1a	70 (12.7%)	17 (12.8%)	
1b	174 (31.5%)	62 (46.6%)	
M, *n* (%) AJCC/UICC 8th edition			
0	525 (95.1%)	110 (82.7%)	<0.001
1	27 (4.9%)	23 (17.3%)	
Stage classification AJCC/UICC 8th edition, *n* (%)			
I	382 (69.1%)	27 (20.3%)	<0.001
II	123 (22.4%)	47 (35.3%)	
III	38 (6.9%)	40 (30.1%)	
IVA	0 (0%)	0 (0%)	
IVB	9 (1.6%)	19 (14.3%)	
Tumor size, *n* (%)			
<4 cm	495 (89.7%)	88 (66.2%)	<0.001
≥4 cm	57 (10.3%)	45 (33.8%)	
Total thyroidectomy, *n* (%)	163 (29.5%)	72 (54.1%)	<0.001
Radioactive iodine therapy, ≥30 mCi, *n* (%)	57 (10.3%)	45 (33.8%)	<0.001

**Table 5 cancers-12-02115-t005:** Cox proportional hazard regression analysis of variables associated with outcome.

Characteristics	Cause-Specific Death			Recurrence		
	HR	95%CI	*p* Value	HR	95%CI	*p* Value
Age ≥55 years	2.47	1.09–6.38	0.03	1.93	1.13–3.35	0.014
Sex	1.51	0.78–2.91	0.216	1.22	0.75–1.97	0.41
T ≥4a	1.3	0.56–3.14	0.55	1.06	0.63–1.79	0.82
Tumor size >4 cm	3.9	1.89–8.42	<0.001	3.72	2.31–5.93	<0.001
N1b	3.27	1.30–8.82	0.011	3.52	2.10–5.95	<0.001
M1	2.46	1.17–5.15	0.018			
*TERT* promoter mutation	5.23	2.33–12.78	<0.001	6.46	3.90–10.83	<0.001

HR, hazard ratio; CI, confidence interval.

**Table 6 cancers-12-02115-t006:** Characteristics of patients in the TERT− group before and after propensity score-matching.

Characteristics	TERT (−)						
	All	Before Matching			After Matching		
		TT	LT	*p* Value	TT	LT	*p* Value
*N*	276	59	217		59	59	
Follow-up duration, y	9.7 ± 3.2	10.2 ± 3.4	9.6 ± 3.0	0.307	10.2 ± 0.4	9.6 ± 0.4	0.439
Age ≥55 years, *n* (%)	115 (41.7%)	33 (55.9%)	82 (33.2%)	0.017	33 (55.9%)	34 (57.6%)	0.645
Sex, female, *n* (%)	234 (84.8%)	51 (86.4%)	183 (84.3%)	0.839	51 (86.4%)	53 (89.8%)	0.777
Tumor dimeter, mm	18.0 ± 5.1	18.1 ± 5.1	18.0 ± 6.8	0.257	18.1 ± 5.1	18.4 ± 5.6	0.944
Bilateral tumor	40 (14.5%)	40 (67.8%)	0 (0%)	<0.001	40 (67.8%)	0 (0%)	<0.001
T3b, *n* (%)	98 (35.5%)	28 (47.5%)	70 (32.3%)	0.033	28 (47.5%)	27 (45.8%)	1.000
Pathological lymph node metastasis, *n* (range)	0.8 ± 1.4 (0–8)	1.0 ± 1.7 (0–8)	0.8 ± 1.4 (0–8)	0.354	1.0 ± 0.7 (0–8)	0.7±1.1 (0–4)	0.355
Outcome							
cause-specific death	1 (1.7%)	1 (1.7%)	0 (0%)	1.000	1 (1.7%)	0 (0%)	1.000
recurrence	6 (2.2%)	1 (1.7%)	5 (2.3%)	1.000	1 (1.7%)	2 (3.4%)	1.000
Complication							
Hypoparathyroidism							
temporary	16 (5.8%)	16 (27.1%)	0 (0%)	<0.001	16 (27.1%)	0 (0%)	<0.001
permanent	5 (1.8%)	5 (8.5%)	0 (0%)	<0.001	5 (8.5%)	0 (0%)	0.057
Recurrent laryngeal nerve paralysis							
temporary	14 (5.1%)	3 (5.1%)	11 (5.1%)	1.000	3 (5.1%)	2 (3.4%)	1.000
permanent	5 (1.8%)	2 (3.4%)	3 (1.4%)	0.291	2 (3.4%)	0 (0%)	0.496

LT, lobectomy; TT, total thyroidectomy, pN, pathologic lymph node.

**Table 7 cancers-12-02115-t007:** Primer sequences and concentrations for PCR.

**(A) For Genomic PCR**				
**Gene**	**Primer name**	**Primer sequence (5′** **→3′)**	**Product size (bp)**	**Concentration (μM)**
*BRAF*	BRAF-int14_5F	GCAGGTTATATAGGCTAAATAGAACTAATC	334	0.2
	BRAF-int15_R	TAGCCTCAATTCTTACCATCCAC		0.2
*TERT* promotor	TERT promotor_1F	CGTCCTGCCCCTTCACCTTC	119	0.1
	TERT promotor_1R	GAAAGGAAGGGGAGGGGCTG		0.1
**(B) For SNaPshot**				
**Target mutation**	**Primer name**	**Primer sequence (5′** **→3′)**	**Concentration (μM)**	
*BRAF V600E*	BRAF_c.1799T > A_F_60mer	ATGCATGCATGCATGCATGCATGCATGCATGCATGCATGGTGATTTTGGTCTAGCTACAG	0.3	
*TERT* promotor C250T	TERT promotor_C250T_30mer	ATGCGCGGACCCCGCCCCGTCCCGACCCCT	0.2	
*TERT* promotor C228T	TERT promotor_C228T_45mer	ATGCATGCATGCATGCATGCATGCCCGGGTCCCCGGCCCAGCCCC	0.2	
**(C) For direct sequencing**				
**Gene**	**Primer name**	**Primer sequence (5′** **→3′)**		
*BRAF*	BRAF-int14_5F	GCAGGTTATATAGGCTAAATAGAACTAATC		
*TERT* promotor	TERT promotor_2F	CACCTTCCAGCTCCGCCTCCTC		

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
