# Peer review of "TERT Promoter Mutation and Extent of Thyroidectomy in Patients with 1–4 cm Intrathyroidal Papillary Carcinoma"

_cancers, 2020, doi:10.3390/cancers12082115_

Round 1

Reviewer 1 Report

Article summary:

In the article “TERT Promoter Mutation and Extent of Thyroidectomy in Patients with 1- to 4-cm Intrathyroidal Papilary Carcinoma”, the Authors aimed to verify whether the BRAF V600E mutation or TERT promoter mutation may be used as predictors of prognosis, and to establish proper indication for lobectomy (LT) in patients with 1- to 4-cm intrathyroidal PTC. It would help in preventing overtreatment nad reduce postoperative complications.

The Authors analysed 685 patients with PTC. For all the patients, clinical data, BRAF V600E mutation status, and TERT promoter status were known. The Authors performed analysis in two steps. In the first step they analysed the whole group of 685 patients. In the second step they concentrated on the subset of 309 patients with small (1-4cm) intrathyroidal PTC.

On the whole group of patients, the Authors compared cause-specific survival (CSS), and disease-free survival (DFS) between samples with different mutation statuses. They showed, that TERT+ samples had worse CSS and DFS then TERT- samples (DFS was analysed after exclusion of M1 samples). They also showed that there was no significant difference in CSS or DFS between BRAF+ and BRAF- samples. They also showed:

  • worse CSS and DFS in BRAF+/TERT+ samples compared to BRAF-/TERT- samples
  • worse CSS and DFS in BRAF-/TERT+ samples compared to BRAF-/TERT- sample
  • no difference in CSS or DFS in BRAF+/TERT- samples compared to BRAF-/TERT- samples

The Authors further compared clinical characteristics of patients between TERT+ and TERT- group, and showed association between TERT promoter mutation and older age, male sex, higher T, higher N, higher M feature, higher stage classification, larger tumor size, and more frequent radioactive iodine therapy.

Further, they concentrated on a subgroup of samples: 276 patients with intrathyroidal PTC with a maximal diameter of 1-4 cm, and no TERT promoter mutation. In that group, only 1 patient died because of cancer, and 6 patients had recurrence (2.2%). The Authors compared the clinical features, outcomes, and complications between patients that underwent total thyroidectomy (TT), and patients that underwent lobectomy (LT). They showed, that TT was more often performed for older patients, bilateral tumors, and T3b. They also showed, that complications were more often in TT group. After propensity score matching (matched by age, sex, tumor size and extrathyroidal invasion of strap muscles), they still showed that TT is more often performed in bilateral tumors, and complications are more often among patients that underwent TT. No difference in frequency of death or recurrence was shown, and no difference in CSS nor DFS according to Kaplan-Meier curves, neither before nor after propensity score matching.

Finally, the Authors analysed a subgroup of 33 patients with intrathyroidal PTC with a maximal diameter of 1-4 cm, and TERT promoter mutation. In this group, there was a large but not statistically significant difference in DFS between patients who underwent lobectomy or total thyroidectomy.

Based on above analysis, the Authors conclude that:

  • TERT promoter mutations were related to poor outcomes for patients in PTC. It has been already shown by other authors, however it was worth to confirm it in Japan population.
  • BRAF V600E was not significantly associated with aggressiveness. This is controversial, but the Authors discuss it in a discussion section.
  • Patients with 1-4 cm intrathyroidal PTC without TERT promoter mutations could obtain favourable outcomes from LT. I think that this is the most important conclusion of the paper.

Comments:

The Authors concluded that BRAF V600E was not significantly associated with aggressiveness. It is not fully consistent with the results. In the results, the Authors showed only that BRAF V600E mutation does not impact CSS nor DFS. But, they did not show, whether it impacts tumor stage, presence of metastasis at the time of diagnosis or tumor size. So, they should not conclude that BRAF V600E mutation is not associated with aggressiveness.

Lines 76, 77

Are the p-values obtained with log-rank test in pairwise comparisons?

Lines 77-79

The Authors wrote, that B-/T+ patients showed significantly worse outcome than B-/T- patients. Could the Authors give the p-values for this comparison? The figure 3 shows only the general p-value for comparison between all 4 groups.

Tables 2 and 4

On figures the cause-specific survival is shown on left and disease-free survival on right. In the table 2 it is inversed. It would be easier for the reader to look at the results, if the cause-specific death is shown on left, and recurrence on right, as on figures.

Line 85

It is not clear, whether multiple univariate models or one multivariate Cox model was performed. Could the Authors clarify it?

Line 93

In line 93 the Authors wrote that 309 patients had intrathyroidal PTC with a maximum diameter of 1-4cm. However, when we look at the table 3, we can see that there were only 102 patients with tumor size below 4 cm. Can the Authors explain the discrepancy?

Table 3

According to text, this table does not describe all TERT- tumors, but only small intrathyroidal TERT- tumors. Can the Authors clarify it in the title of the table?

Table 5

Please recheck and correct the table 5 as there are multiple small errors:

  • There is a row “Follow-up duration, y (range)” but no range is shown.
  • There is also a row “tumor size, n(%)”, but the tumor size is expressed with mean and SD.
  • Please, clarify what the tumor size is. Is it a tumor dimeter in mm?
  • Please, clarify, what the pN mean. Is it a number of pathologic lymph nodes?
  • There is one cause-specific death among TT patients, but no cause-specific death among all patients?

Figures 4, and 5

As I understand, the figures show survival analysis for patients with intrathyroidal PTC with a maximal diameter of 1-4cm and no TERT promoter mutation. Could the Authors add this information to the figure title?

Author Response

Reviewer #1 comment

The Authors concluded that BRAF V600E was not significantly associated with aggressiveness. It is not fully consistent with the results. In the results, the Authors showed only that BRAF V600E mutation does not impact CSS nor DFS. But, they did not show, whether it impacts tumor stage, presence of metastasis at the time of diagnosis or tumor size. So, they should not conclude that BRAF V600E mutation is not associated with aggressiveness.

Ans)

Thank you very much for your remark. Accordingly, we analyzed the patients’ characteristics with and without BRAF V600E mutation in all cases and added Table 3. BRAF V600E mutation was significantly more common among older patients (p<0.001) and cases with higher T and stage at diagnosis (p<0.001); however, it was inversely associated with higher N and M. Thereafter, BRAF V600E mutation did not impact on poor CSS and DFS. Thus, we concluded that BRAF V600E mutation alone was not significantly associated with aggressiveness in this series.

Lines 76, 77

Are the p-values obtained with log-rank test in pairwise comparisons?

Lines 77-79

The Authors wrote, that B-/T+ patients showed significantly worse outcome than B-/T- patients. Could the Authors give the p-values for this comparison?

Ans)

Yes. We calculated these p-values using the log-rank test in pairwise comparisons and have noted this at 4.4 Statistical analysis section in Materials and Methods.

We added these p-values in Line 78-80.

The figure 3 shows only the general p-value for comparison between all 4 groups.

Ans)

As per your suggestion, we revised figure 3 and added all p-values for 6 patterns of combination.

Tables 2 and 4

On figures the cause-specific survival is shown on left and disease-free survival on right. In the table 2 it is inversed. It would be easier for the reader to look at the results, if the cause-specific death is shown on left, and recurrence on right, as on figures.

Ans)

We appreciate for your suggestion. We revised Table 2 and 4; cause-specific death is set on left and recurrence is set on right.

Line 85

It is not clear, whether multiple univariate models or one multivariate Cox model was performed. Could the Authors clarify it?

Ans)

We performed multivariate Cox proportional hazard regression analysis. Then, we described clearly this in Line 80 and 88.

Line 93

In line 93 the Authors wrote that 309 patients had intrathyroidal PTC with a maximum diameter of 1-4cm. However, when we look at the table 3, we can see that there were only 102 patients with tumor size below 4 cm. Can the Authors explain the discrepancy?

Ans)

Thank you for your remark. We wrote the item <4 cm and ≥4 cm in reverse order on Table 1 and 4 by mistake. There were 583 patients with tumor size <4 cm in total and 309 patients with 1- to 4-cm intrathyroidal PTC.

Table 3

According to text, this table does not describe all TERT tumors, but only small intrathyroidal TERT- tumors. Can the Authors clarify it in the title of the table?

Ans)

Table 3 and 4 included all the 685 patients.

Table 5

Please recheck and correct the table 5 as there are multiple small errors: There is a row “Follow-up duration, y (range)” but no range is shown.

There is also a row “tumor size, n(%)”, but the tumor size is expressed with mean and SD.

Please, clarify what the tumor size is. Is it a tumor dimeter in mm?

Please, clarify, what the pN mean. Is it a number of pathologic lymph nodes?

There is one cause-specific death among TT patients, but no cause-specific death among all patients?

Ans)

Thank you very much for your kind remarks. We corrected those errors in revised Table 6.

Figures 4, and 5

As I understand, the figures show survival analysis for patients with intrathyroidal PTC with a maximal diameter of 1-4cm and no TERT promoter mutation. Could the Authors add

this information to the figure title?

Ans)

Thank you for your suggestions. We added the information about target patients in the figure 4, 5 and 6.

Reviewer 2 Report

The paper provides a very interesting observation that the presence of TERT promoter mutation in PTC could serve as a useful marker in qualification for TT. Taking into account the differences in the surgical approach in most European countries and in Japan, such data can broaden our knowledge and constitute the basis for multicentre prospective studies, the results of which may influence current guidelines.

However, some additional information should be provided.

                The authors analysed only mutations of BRAFV600E and TERT promoter. It is obvious that these mutations are common in PTC and were reported as markers of worse prognosis. But still I do not fully understand why no other PTC-related mutations were studied in such a large group of patients. The frequencies of RET/PTC rearrangements or RAS mutations in PTC are assessed as 6-30% and 10-20%, respectively (not mentioning rarer mutation, such as TRK or others) (J. Clin. Med. 2019, 8, 1916). The authors compared the analysed patients to the patients with “neither mutation” but we do not know what “neither mutation” really meant, as several not analysed genetic alterations might have been included in that group. Please explain why the study included only the two genes mutations and discuss a potential influence of such approach on the obtained results.

                Additionally, it was widely reported that the coexistence of  BRAF V600E and TERT promoter mutations are correlated with more aggressive PTC than the presence of TERT mutation only. Your study provided different data (e.g. Fig. 3, Table 2). Please further explain this phenomenon in the discussion.

                Moreover, the conclusion provided in your study can be useful for adult patients only. In children, TERT promoter mutations are practically not observed in PTC but LB cannot be applied in most paediatric cases. Please refer to this issue to avoid misunderstanding and possible undertreatment of  children with PTC.

Author Response

Reviewer #2 comment

However, some additional information should be provided.

The authors analysed only mutations of BRAFV600E and TERT promoter. It is obvious that these mutations are common in PTC and were reported as markers of worse prognosis. But still I do not fully understand why no other PTC-related mutations were studied in such a large group of patients. The frequencies of RET/PTC rearrangements or RAS mutations in PTC are assessed as 6-30% and 10-20%, respectively (not mentioning rarer mutation, such as TRK or others) (J. Clin. Med. 2019, 8, 1916). The authors compared the analysed patients to the patients with “neither mutation” but we do not know what “neither mutation” really meant, as several not analysed genetic alterations might have been included in that group. Please explain why the study included only the two genes mutations and discuss a potential influence of such approach on the obtained results.

Ans)

Thank you very much for your remarks. Actually, we analyzed other genetic alterations including RAS mutation, RET/PTC rearrangements and TRK fusions using the same samples as this study. And we are preparing another paper studying the frequency of those mutations and their impacts on treatment outcomes in this Japanese population.

In this paper, we aimed to reveal appropriate indication of lobectomy in patients with 1- to 4-cm intrathyroidal PTC using the genetic status of the tumor to avoid overtreatment. For this purpose, we picked up BRAF and TERT as the most popular mutations affecting the prognosis of PTC.

We clarified “neither mutation” meant “B-/T-“in Line 75-76 and thereafter.

Additionally, it was widely reported that the coexistence of BRAF V600E and TERT promoter mutations are correlated with more aggressive PTC than the presence of TERT mutation only. Your study provided different data (e.g. Fig. 3, Table 2). Please further explain this phenomenon in the discussion.

Ans)

You are totally right and we have described in Line 226-230 that coexisting BRAF V600E and TERT promoter mutations represented a strong predictor for the most aggressive PTCs with the highest recurrence rate. Moreover, some meta-analyses have verified that coexistence of both mutations has a synergistic effect on aggressive clinicopathological characteristics and even cancer-related mortality for patients with PTC.

In our series, although coexisting BRAF V600E and TERT promoter mutations were independently related to CSS for patients with PTC; the majority of the patients (78.5%) had BRAF mutation and most (126 of 133) patients with TERT promoter mutations also had BRAF mutation. Thus, BRAF V600E mutation was not stronger indicator associated with CSS and DFS compared to TERT mutations in this Japanese population. Further study would be needed to clarify the reason why Japanese patients showed higher incidence of BRAF mutation than other countries.

We added these comments in Line 220-223 and 235-238.

Moreover, the conclusion provided in your study can be useful for adult patients only. In children, TERT promoter mutations are practically not observed in PTC but LB cannot be applied in most paediatric cases. Please refer to this issue to avoid misunderstanding and possible undertreatment of children with PTC.

Ans)

Thank you very much for an important remark. We added this point to the end of Discussion (Line 249-251).

Reviewer 3 Report

The present study focused to questioned any genetic mutations as predictors of prognosis to be able to establish proper indications for lobectomy (LT) in patients with 1- to 4-cm intra-thyroidal PTC in a prospectively collected 685 consecutive patients. 

The study has nicely designed and underline very important hot topic in the thyroid literature.

My question is what the authors were defined as "low-risk" and "high-risk" were defined based on as clinical characteristics. However, sometimes even the clinically high-risk characteristics may not exist but the tumour may show the morphology of aggressive variants of PTC. How was the ratio of  aggressive variants of PTC in TERT+ and TERT - group? There are also papers that have started to show up underlining the presence of TERT mutation in benign thyroid neoplasms (https://doi.org/10.3390/cancers12071846 ). It would be great if the authors can add information about it.

Author Response

Reviewer #3 comment

My question is what the authors were defined as "low-risk" and "high-risk" were defined based on as clinical characteristics. However, sometimes even the clinically high-risk characteristics may not exist but the tumour may show the morphology of aggressive variants of PTC. How was the ratio of aggressive variants of PTC in TERT+ and TERT - group?

Ans)

Thank you very much for your comment. We classified patients into “high-risk” and “low-risk” group according to our definition (reference number 13) using clinical characteristics provided before or during initial surgery.

One of the purposes of this study is to reveal appropriate indication of lobectomy for patients with 1- to 4-cm intrathyroidal PTC using the genetic status which could be revealed before surgery. Thus, we selected genetic mutations that we could examine by cytology as new diagnosis tools.

You are totally right that pathological types of PTC would be an important factor affecting the treatment outcomes. However, aggressive variants of PTC like tall-cell variant (TCV) are seldom determined by cytology only. Moreover, we encountered only one case of TCV in this series. Thus, we omitted the information from the study.

There are also papers that have started to show up underlining the presence of TERT mutation in benign thyroid neoplasms (https://doi.org/10.3390/cancers12071846 ). It would be great if the authors can add information about it.

Ans)

Thank you for the remark. We added this literature as Ref. 28.

Round 2

Reviewer 1 Report

I accept all the responses and corrections made by Authors.